# Importance of the Immune Microenvironment in the Spontaneous Regression of Cervical Squamous Intraepithelial Lesions (cSIL) and Implications for Immunotherapy

**DOI:** 10.3390/jcm11051432

**Published:** 2022-03-05

**Authors:** Caroline L. P. Muntinga, Peggy J. de Vos van Steenwijk, Ruud L. M. Bekkers, Edith M. G. van Esch

**Affiliations:** 1Department of Gynecology and Obstetrics, Catharina Ziekenhuis Eindhoven, Michelangelolaan 2, 5623 EJ Eindhoven, The Netherlands; caroline.muntinga@catharinaziekenhuis.nl (C.L.P.M.); ruud.bekkers@catharinaziekenhuis.nl (R.L.M.B.); 2GROW—School for Oncology and Reproduction, Maastricht University, Universiteitssingel 40, 6229 ER Maastricht, The Netherlands; peggy.de.vosvansteenwijk@mumc.nl; 3Department of Gynecology and Obstetrics, Maastricht Universitair Medisch Centrum, P. Debyelaan 25, 6229 HX Maastricht, The Netherlands

**Keywords:** cervical high-grade squamous intraepithelial lesions, immune microenvironment, immunology, human papillomavirus, spontaneous regression, immunotherapy, imiquimod

## Abstract

Cervical high-grade squamous intraepithelial lesions (cHSILs) develop as a result of a persistent high-risk human papilloma virus (hrHPV) infection. The natural course of cHSIL is hard to predict, depending on a multitude of viral, clinical, and immunological factors. Local immunity is pivotal in the pathogenesis, spontaneous regression, and progression of cervical dysplasia; however, the underlying mechanisms are unknown. The aim of this review is to outline the changes in the immune microenvironment in spontaneous regression, persistence, and responses to (immuno)therapy. In lesion persistence and progression, the immune microenvironment of cHSIL is characterized by a lack of intraepithelial CD3+, CD4+, and CD8+ T cell infiltrates and Langerhans cells compared to the normal epithelium and by an increased number of CD25+FoxP3+ regulatory T cells (Tregs) and CD163+ M2 macrophages. Spontaneous regression is characterized by low numbers of Tregs, more intraepithelial CD8+ T cells, and a high CD4+/CD25+ T cell ratio. A ‘hot’ immune microenvironment appears to be essential for spontaneous regression of cHSIL. Moreover, immunotherapy, such as imiquimod and therapeutic HPV vaccination, may enhance a preexisting pro-inflammatory immune environment contributing to lesion regression. The preexisting immune composition may reflect the potential for lesion regression, leading to a possible immune biomarker for immunotherapy in cHSILs.

## 1. Introduction

Cervical cancer is preceded by premalignant stages known as cervical high-grade squamous intraepithelial lesions (cHSILs), also referred to as cervical intraepithelial neoplasia (CIN) 2 and CIN 3, respectively, moderate and severe dysplasia [1,2]. cHSIL is the most commonly known premalignant neoplasia, affecting approximately 1–2% of all women worldwide each year [3]. Development of these lesions is causally related to a persistent infection with high risk human papillomavirus (hrHPV) [4,5,6,7]. Human papilloma virus (HPV) infections are the most common sexually transmitted pathogens worldwide with an estimated life-time risk of 80% [4,5,6]. While about 80% of hrHPV infections are cleared by the immune system within 18 months, a small proportion of women fail to control viral infection [5,8,9]. Persistent hrHPV infections develop in approximately 10% of the infected women and are associated with the development of cervical low-grade squamous intraepithelial lesions (cLSILs), cHSILs, and their subsequent progression into invasive squamous cell carcinoma [4,5,6]. The majority of cHSILs and cervical cancer are caused by a persistent infection with oncogenic hrHPV types, in particular hrHPV16 and hrHPV18 [10,11,12,13]. In cHSIL, hrHPV infected cells in the epithelium express the early viral oncogenes E6 and E7, which are key in the onset, maintenance, and progression of the lesion by dysregulation of the cell cycle [14,15,16]. In the natural course of the disease, spontaneous regression is reported, especially in cLSILs, where 60% of lesions regress without action [17]. The natural course of cHSIL, however, is different, with only approximately 25% of lesions regressing spontaneously, while up to 18% of lesions will progress into invasive cervical cancer in the long-term if left untreated [3,17,18]. 

Since cHSILs have a clear malignant potential, screening and therapies are aimed at the prevention of progression into invasive carcinomas. A distinction in the treatment approach can be made between CIN 2 and CIN 3; in Table 1, an overview of the treatment strategies in different countries is given. Women with cLSILs and women under 30 years with CIN 2 can be managed expectantly, with adequate cytological follow-up, since regression rates up to 60% after 24 months have been described in both groups [17,18]. CIN 3 lesions are, in general, effectively treated with a large loop excision of transformation zone (LLETZ), with success rates of approximately 90% [3,18]. LLETZ is effective, however, it is an invasive procedure, with a risk of complications and side effects. The most serious side effects are premature birth due to cervical insufficiency and subfertility [19,20,21]. The risk of overall preterm birth (less than 37 weeks gestational age) increases from 5.4% to 10.7% and the risk of severe prematurity (less than 32 to 34 weeks) more than doubles from 1.4% to 3.5% after excisional therapy compared to untreated women. Moreover, a positive correlation between preterm birth and higher excised volumes of cervical tissue has been described [19,22]. A relatively new, non-surgical treatment of cHSIL is imiquimod with clinical responses up to 60–73% [23,24]. Imiquimod is an immune modulating cream and a good alternative therapy for cHSIL, especially in women with (possible) future pregnancy wishes. The side effects, however, upon imiquimod application, are common, and can be extensive, including systemic adverse events, but they mostly consist of local inflammation and burning [25]. Consequently, therapy adherence is challenging, with up to 20% discontinuation of treatment due to the side effects and treatment duration of 16 weeks [26,27]. Therefore prediction of spontaneous regression and imiquimod effectivity in cHSIL would be of aid in patient selection and counseling. At present, no predictive (bio)markers are known for predicting cLSIL or cHSIL regression or progression. 

The immune system is known to play an important role in the protection against hrHPV infection and the regression and persistence of hrHPV-induced pathology, as reflected by the increased risk of HPV-related lesions and (pre)malignancies in immunosuppressed patients, e.g., HIV positive or transplant patients [8,33,34]. The importance of an adaptive pro-inflammatory HPV-specific immune response to clear a hrHPV infection is reflected by the presence of a strong HPV-specific pro-inflammatory systemic immune reaction in 60% of healthy individuals, in contrast to only 30% of patients with CIN or cervical cancer. Moreover, the HPV-specific immune reaction in CIN or cervical cancer is characterized by anti-inflammatory HPV-specific immune responses, which promote tumor tolerance [35,36]. Viral-based factors play an important role in the evasion of the hosts immune system and, subsequently, in lesion persistence. HPV is able to induce immune evading mechanisms in HPV-infected keratinocytes, leading to cell cycle dysregulation, chromosomal instability and resistance to apoptosis [37]. For example, E7 can inhibit STING pathway activation, which stimulates apoptosis and senescence of dysplastic cells and enhances the effect of cytotoxic T cells [38]. Other reports on the E7 oncoprotein state that E7 downregulates cell-surface MHC-I molecules, IFN-α and interferon-regulatory factor-1 (IRF-1), potentially leading to a reduction in the presentation of viral antigens, subsequently leading to a delayed reaction from the adaptive immune system [39,40]. In the spontaneous regression of lesions, the adaptive immune system is able to identify E6 and E7 antigens in the HPV-infected keratinocytes and to clear the infection [40,41,42]. In a large prospective study of HPV16+ cLSIL, systemic T-cell responses to HPV16 E2 protein were associated with regression, while T-cell responses to HPV16 E6 protein are correlated to persistence [43]. Since hrHPV infections are primarily localized in the epithelium, the local immune response in the microenvironment of cHSIL is essential in the first line of defense. Remarkably, the immune microenvironment in cervical HSIL has not been studied in depth in relation to spontaneous regression and responses to (immuno)therapy. Scarce studies, with limited patient numbers describe the type, numbers, and distribution of infiltrating immune cells; however, in depth studies are lacking. 

Since the immune system plays an essential role in the natural history of HPV-related lesion development, persistence, and clearance, the aim of this narrative review is to provide an outline of the changes in the immune microenvironment of premalignant cervical lesions to provide insight into the importance of the local immune microenvironment in both cLSIL and cHSIL for spontaneous regression and (immuno)therapy responses. A methodological search on PubMed was performed, resulting in 697 results (Appendix A), of which, we describe the most relevant findings. Cross references were included. 

## 2. The Importance of Immune Infiltrates in the Microenvironment of cLSIL and cHSIL in the Natural Course of the Disease

The immune microenvironment in cervical neoplasia is dynamic and changes over time when lesions regress or progress into cHSIL or cervical cancer. The microenvironment consists of a multitude of cell types, including infected keratinocytes, immune cells, endothelial cells, pericytes, mesenchymal stem cells and fibroblastic cells, which are all intrinsically related to each other [44]. There are, however, pivotal differences in the various immune cells within the microenvironment of healthy cervix, cLSIL, and cHSIL, which we will describe below. The reported changes are summarized in Table 2. A visual overview is presented in Figure 1. 

### 2.1. Innate Immune Responses in cLSIL and cHSIL

The innate immune system has an important function as first line of defense and is able to respond immediately and non-specific upon pathogen encounter. The innate immune system has an important, yet contradictory role in tumor regulation, being able to simultaneously prevent tumor progression via antigen-presenting cells (APCs) and type 1 (M1) macrophages, and aid tumor growth via myeloid-derived suppressor cells and type 2 (M2) macrophages [66,67]. We will first outline the role of professional APCs, which are crucial in adaptive anti-tumor responses since these connect the innate and adaptive immune system; we then discuss macrophages in cervical neoplasia (e.g., cLSIL and cHSIL).

#### 2.1.1. Antigen Presenting Cells

Antigen presenting cells (APCs), e.g., dendritic cells (DCs) and Langerhans cells (LCs), are the link between the innate and adaptive immune response by their ability to prime naïve T cells in secondary lymphoid organs, such as lymph nodes [68]. Immature DC’s (iDC) need to be correctly activated and maturated in order to adequately present antigens to B- and T cells leading to the activation of naïve CD4+ and CD8+ T cells and the secretion of pro-inflammatory cytokines. HPV may prevent the activation and maturation of dendritic cells by its immune evading mechanisms, leading to inadequate antigen presentation and subsequently inhibiting an effective pro-inflammatory T cell response [69]. iDCs are increased in cHSIL compared to normal cervical tissue [45]. iDCs facilitate T cell anergy and are tolerant to dysplastic cells by producing immunosuppressive enzyme indoleamine 2,3-dioxygenase (IDO)-1, transforming growth factor (TGF)-β and Interleukin (IL)-10 [62]. Due to IL-10 and TGF-β, the differentiation of DCs is inadequate, which may lead to regulatory T cell (Treg) induction. Moreover, IDO-1 can cause suppression of cytotoxic T cells by tryptophan depletion [45,70]. In cHSIL there is an increased expression of IDO-1 by cervical epithelial cells and macrophages which leads to peripheral tolerance to high risk-HPV transformed cells and T cell anergy [45,56,61]. In disease progression from normal epithelium to cHSIL, the expression of co-stimulatory molecules CD80 and CD86 on DCs is decreased and the expression of co-inhibitory programmed death-ligand 1 (PD-L1) increased, implicating impaired T cell activation by DCs [60]. The expression of PD-L1 in dysplastic squamous cells of the epithelium is increased in cHSIL up to 95% compared to no PD-L1 expression in normal epithelium [59]. Upon programmed death (PD)-1/PD-L1 interaction T cell proliferation and activity of cytotoxic T cells may be impaired [59]. In addition, reduced levels of Th1 cytokines IFNy and IL-12 and an increased level of Th2 cytokine IL-10 were measured during hrHPV positivity and increasing grades of dysplasia [60]. In cHSIL, the overall amounts of both epithelial CD1a+ LCs and epithelial HLA-DR+ LCs are reduced compared to cLSIL and healthy cervical tissue [46,47,48,49,50]. Interestingly, in cLSIL, no reduced numbers of epithelial LCs are reported [46]. Moreover, the LCs in both cLSIL and cHSIL expressed significantly more HLA-DQ compared to healthy cervix, indicating activation and maturation and the ability to present antigens to CD4+ T cells, in order to initiate a cell-mediated immune response via major histocompatibility complex (MHC) class II to induce lesion regression [46,48,71]. Clearance of hrHPV infection is associated with increased epithelial LC infiltration [72]. LC depletion in cHSIL epithelium may be either caused by an increased migration of LCs from the epithelium to the lymph nodes, where they present viral or tumor associated antigens to lymphocytes, or by impaired function of LCs. Vaccination with HPV virus-like particles (VLP), both L1L2 VLP and chimeric L1L2E7 VLP, showed that LCs are not activated upon interaction with VLP and do not migrate out of the epidermis in a mouse model, these data demonstrate that LCs are unable to mount a HPV-specific T cell response [49,73]. The depletion of LCs may cause local immune suppression by reduced antigen presentation, hereby resulting in an impaired cell-mediated immune response. Furthermore, in CIN 2 lesions, expression of mRNA for toll-like receptor (TLR) 2, 3, and 7–9 is significantly increased in patients who cleared the HPV infection, whereas reduced TLR expression was observed in persistent CIN 2 lesions [74,75,76]. TLRs are essential in antiviral responses since they trigger a pro-inflammatory response after activation of the NF-kB pathway. Moreover TLR-agonists can be used to restore immunogenicity in tumors lacking the TLR-ligand [77]. 

#### 2.1.2. Macrophages

Macrophages are, in general, divided into two types; type 1 macrophages (M1), which have a pro-inflammatory effect, and type 2 macrophages (M2), which are anti-inflammatory and are associated with lesion progression [67,78]. Both groups express CD68, while CD163 is often expressed by M2 macrophages [79]. A dense infiltration of M1 macrophages in HPV-associated cervical cancer is an independent prognostic factor in survival [78], whilst M2 macrophages can promote tumor metastasis, suppress T cells in the tumor microenvironment and are correlated with higher cervical cancer stages [53,66,80]. 

In cervical dysplasia, an increase in intraepithelial CD68+ macrophage infiltration is seen in the course of lesion progression to cervical cancer [45,51,52,53]. Persistent cLSIL is characterized by a higher intraepithelial CD68+ macrophage count compared to spontaneous regressing cLSIL [52]. A significant increase in CD163+ cells, reflecting M2 macrophages, was observed in the epithelium of cHSIL during disease progression compared to cLSIL and normal cervix [53]. During disease progression, both CD68+ and CD163+ cell infiltrates increased in the epithelium, with a more distinct increase in intraepithelial CD163+ cells, indicating that macrophages, and M2 macrophages in particular are recruited into the epithelium [53]. Subsequently, macrophage activation by HPV-infected tumor cells in the immune microenvironment may result in lower antigen presenting ability and suppressed T cell proliferation both aiding lesion progression [52,81].

### 2.2. Adaptive Immune Responses in cLSIL and cHSIL

The adaptive immune system comprises two different immune reactions, humoral immunity, which is mediated by antibodies produced by B lymphocytes, and cell-mediated immunity, facilitated by T lymphocytes, which react against intracellular replicating organisms, such as HPV [54,82]. The adaptive immune system can recognize the E6 and E7 antigens and eliminate the HPV infected cells [54,83]. In persistent HPV infections, various immune escape mechanisms in the infected cell prevent an adequate HPV-specific T-cell mediated immune response [15,83]. 

#### 2.2.1. T Cells

In cLSIL, lower counts of CD4+ and CD8+ T cells were observed compared to healthy cervical tissue in both stromal and epithelial compartments, as presented in Table 2 [39,54,55]. In cHSIL, a decreasing trend in the amount of CD4+ and CD8+ cells compared to normalcy and cLSIL is likewise observed in single cell counts via flowcytometry by Wang et al. [56]. CD4+ rates were notably higher in cLSIL compared to cHSIL. However, a relatively larger reduction in epithelial CD4+ T cells was observed in both cLSIL and cHSIL compared to the stromal compartment [37,54]. Interestingly, the regressors in a cLSIL cohort showed a higher intraepithelial CD8/CD4+ T cell infiltrate ratio, indicating the importance of preexistent CD8+ intraepithelial infiltrates in the natural course of cLSIL [37]. Interestingly, the ratio between epithelial CD8+ and granzyme B+ cells was close to one, which indicates that the infiltrated CD8+ T cells in cLSIL are highly active and able to initiate lesion regression through multiple pathways [37,84]. Granzyme B is only expressed in the cytoplasm of activated CD8+ cytotoxic T cells and induces apoptosis in virus infected cells [37]. In the group with progressive/persistent cLSIL, this CD8+/granzyme B ratio was three fold lower indicating a diminished amount of effector CD8+ T cells necessary to clear cLSIL [37].

In the natural course of disease in cHSIL, different associations with intralesional T cell infiltrates are described in either lesion regression, progression, persistence, or recurrence [54]. A recently published meta-analysis on T cell infiltrates in cervical carcinogenesis included 73 studies and concluded that the overall amount of CD3+, CD4+, and CD8+ T cell infiltrates in cHSIL is reduced compared to normal cervical tissue [54]. Low expression of stromal CD138+ B cells, high expression of stromal CD8+ and high ratios of epithelial CD4+/CD25+ T cells are reported predictors for spontaneous regression in cHSIL [83]. In general, the nine studies on T cell infiltrates, which are limited by single stain immunohistochemistry in small study cohorts, suggest that an increased number of CD4+ and CD8+ T cells in cHSILs are associated with improved outcomes, while no distinction in localization of these T cells is given [54]. Trimble et al. showed that dysplastic lesions with high counts of epithelial CD8+ T cells were 22 times more likely to regress and in persistent cHSIL, CD8+ cells were limited to the stroma [85]. The improved outcomes of dense CD4+ and CD8+ T cell infiltrates have also been described for HPV positive oropharyngeal squamous cell carcinoma and vulvar squamous cell carcinoma [86,87]. 

A recent study of Wang et al., based on single cell suspensions of fresh cervical tissues where flow cytometry and gene expression profiles between different pathological stages of cervical dysplasia were analyzed, marks the importance of an in depth analysis where from cLSIL onwards immune surveillance is unleashed and immune-suppression mechanisms are triggered in cHSIL [56]. An increase in B cells, total T cells, Tregs, monocytes, neutrophils, and M2 macrophages from normal cervix to cancer is seen and they report a decrease in CD4+ T cells and CD8+ T cells. Differently expressed genes in cHSIL were involved in the activation pathways during lesion progression, such as cell proliferation, DNA damage, immune response, metabolism, and unfolded protein response pathway. Epithelial–mesenchymal transition lacked in cHSIL [56]. Furthermore MHC-II molecules were upregulated on CD4+ T cells from normalcy to cHSIL triggered by the increased production of pro-inflammatory cytokines [56]. 

#### 2.2.2. Immune Evading Mechanisms during Disease Progression and Persistence

Checkpoint expression increased with disease progression in cHSIL, shown by increased numbers of CD8+CTLA-4+, CD8+IDO+ and CD8+PD-1+ T cells; however, the numbers of CD8+TIM3+ T cells did not increase towards cHSIL [56]. Other studies observed high numbers of PD-1 T cells and PD-L1 expression compared to normal cervical tissue, moreover higher prevalence of PD-1+ T cells is associated with persistence [59,60,88]. PD-1 is expressed on active T cells. By binding with its ligand, PD-L1, which can be expressed on dysplastic epithelial cells and APCs, a tolerogenic environment is created through a reduced T-effector response and differentiation of naïve T cells into Tregs [88,89,90]. The amount of stromal CD25+FoxP3+ Tregs is increased in cLSIL and even more pronounced cHSIL, compared to healthy cervix [37,45,54,56,57,58]. In addition, higher amounts of CD25+ and Foxp3+ Tregs were observed in persistent cervical dysplasia [56,74,83,88]. This might be explained by the ability of Tregs to suppress the specific cell-mediated immune reaction to HPV infected cells, leading to lesion persistence and even progression [83,91]. A shift from Th1 to Th2 cytokines is observed in cHSIL leading to suppressive mechanisms such as immature DCs, Treg induction and polarization of M2 macrophages [56]. Higher levels of Tregs produce immunosuppressive cytokines, such as TGF-β, IL-10 and IL-35 and are associated with more severe disease and significantly reduced overall survival in cervical carcinomas [55,63,64,65]. TGF-β prevents T cell infiltration into dysplastic tissue and T cell activation [92,93]. IL-35 causes local CD4+ and CD8+ T cell depletion and IL-10 suppresses the production of pro-inflammatory cytokines and prevents antigen presentation by APCs [55,61,94,95]. Peghini et al., consistent with this finding, observed an increase in Treg associated cytokine profile in cHSIL compared to cLSIL [96]. This is in line with the observed enrichment of Tregs in persistent HPV induced cervical neoplasia [97]. Besides Tregs, M2 macrophages, APCs, and epithelial cells also produce anti-inflammatory cytokines, which contribute to an immune evading microenvironment [55,94].

In summary, lower counts of infiltrating CD4+ and CD8+ T cells in cHSIL indicate an evasion of the adaptive immune system by a persistent hrHPV infection, what might lead to lesion persistence and progression into cervical cancer. Furthermore, upon PD-1/PD-L1 interaction reported in cHSIL with a higher percentage of PD-1+ T cells observed compared to normal epithelium, T cell tolerance may be induced and may be responsible for the suppressed immune microenvironment associated with disease progression [56,60,88]. As illustrated in Figure 1, when cHSIL regresses, the dysplastic cells are replaced by healthy cells developing from basal cells. Nevertheless, these healthy cells may still contain the HPV-genome, without expressing their viral genes. Subsequently, in the case of a change in immune status, viral re-activation of latency may occur and cause recurrent disease [98,99]. One study, which evaluated recurrence after LLETZ, showed that the amount of stromal and epithelial CD4+ and T-bet+ cells and stromal CD11c+ cells present at time of excision were correlated with a significantly reduced recurrence rate [69]. 

In conclusion, when comparing the immune microenvironment of cLSIL to cHSIL, lower numbers of epithelial LCs, epithelial, and stromal CD4+ and CD8+ T cells are reported in cHSIL, and an increase in the number of macrophages, mainly epithelial M2 macrophages, and stromal Tregs, PD-L1+, and IDO-1 expression was observed, associated with a tumor tolerant cytokine profile. Thus, during lesion progression from cLSIL to cHSIL, the tumor immune microenvironment develops into a more immunosuppressed anti-inflammatory environment. An effective cell-mediated immune response appears to be necessary to clear the infection, a preexistent so called ‘hot’ immune microenvironment, is pivotal for spontaneous regression. 

## 3. Importance of Clinical Parameters and HPV Genotype in the Natural History of Disease in Relation to the Immune System

Spontaneous regression occurs in up to 60% of cLSIL lesions, whereas in cHSIL, regression is observed in only 25 to 40% of lesions [17,83,100]. cHSIL lesions however can be subdivided into CIN 2 and CIN 3 lesions. Since the majority of the CIN 3 lesions will persist or progress when left untreated, all lesions irrespective of age are treated [18]. More prudence can be taken with CIN 2 lesions, where surveillance should be discussed if possible, especially in women under 30 years old [17,18,101,102]. As described before, multiple immunologic parameters influence the local immune microenvironment of cHSIL, which can contribute to lesion regression or persistence. Moreover, different clinical parameters are also known to influence the probability of spontaneous regression in cervical neoplasia and, therefore, can aid the clinician in the treatment decision-making process [103]. We will discuss the known role of the immune system in relationship to these clinical parameters. 

Age is an important clinical parameter since women under 30 years old with cHSIL, especially with CIN 2, are more prone to spontaneous regression than older women [101,104,105,106]. In women under 30 years CIN 2 has spontaneous regression rates of up to 66%, compared to 50% in women above 30 years, and thus CIN 2 can be managed conservatively especially in women under 30 years old [17,18,101,102]. For CIN 3 patients, age is of no influence when making a decision for treatment, considering overall higher progression rates of 2% and hence treatment is indicated at all ages. In aging the immune system functions decline, which is called ‘immunosenescence’ [107,108]. This already starts at sexual maturity, which might be an explanation for the significantly higher percentage of spontaneous regression in cHSIL in women under 30 [17,108]. Moreover a retrospective cohort study reports that in younger patients the prevalence of hrHPV genotypes is higher compared to women >50 years before treatment whereas HPV persistence and recurrence after treatment is higher with increasing age [109]. Immunosenescence may be an important determinant of hrHPV and cHSIL persistence [105], however studies comparing the systemic and local immunity in cHSIL lesions at different ages are lacking.

Smoking increases the risk of cHSIL, disease persistence, and progression and HPV persistent, even after surgical treatment [103,104,110,111,112]. Smoking causes a local immunosuppressive environment at the cervix, since there are decreased amounts of HLA-DR+ and CD1a+ LCs and CD4+ T cells, which is correlated to persistent cervical HPV infection [113,114]. No changes in CD25+FoxP3+ or PD-1+ T cells were observed in smoking women with cLSIL and cHSIL [88]. Consistent condom use was associated with a protective effect on HPV infections in a systematic review of ten articles [115]. A significantly higher regression rate was observed in patients with cHSIL who used condoms consistently, compared to patients with cHSIL who did not use condoms [116,117,118]. This effect might be mediated by lower repetitively exposure of the cervical tissue to HPV and semen, which has an immunosuppressive effect [118]. Lastly, nulliparous women with cHSIL have a 5 times greater likelihood of spontaneous regression than parous women [119]. This might be explained by increased hormone levels and weakened immune response during pregnancy and tissue damage of the cervix during vaginal delivery [120,121]. 

HPV genotyping may be of importance in lesion regression. In the study of Øvestad et al., no regression was observed in HPV16-containing lesions. This is in line with other evidence showing that HPV16-positive cHSIL regress less often, which might be explained by early integration of HPV16 into the host cell DNA [83,100,122,123,124]. 

The vaginal microbiota (VMB) also seem to play a role in the acquisition and persistence of HPV, as well as in the clearance of cHSIL [125]. An ecosystem with reduced amounts of lactobacilli and a great diversity of bacterial species often associates with vaginal dysbiosis, such as bacterial vaginosis (BV) [126,127,128]. Epidemiological studies have revealed associations between the *Lactobacillus*-dominant vaginal microbiota and decreased detection of HPV infection and dysplasia [129]. Conversely, the overall depletion of *Lactobacillus* species and overgrowth of anaerobe bacteria, have been associated with HPV acquisition and persistence [129]. Similarly, studies showed a depletion of *Lactobacillus* spp. and increase in microbiota’s diversity in cHSIL and cervical cancer [129]. 

The mechanisms of interaction between the HPV, the VMB and host’s immune response are currently largely unclear, however there are indications that the VMB influences the host’s vaginal and cervical microenvironment and its immune response to pathogens [83,130,131].

Lactobacilli play an important role in the defense to pathogens through their ability to produce lactic acid and maintain a low pH, which is harmful to most pathogens [132]. In contrast, vaginal dysbiosis is associated with an increase in pH, the induction of local inflammation and release of chemotactic mediators [133]. Indeed, the increased microbial diversity typical of vaginal dysbiotic states, is associated with the production of pro-inflammatory cytokines and chemokines, as was shown in women with low abundance of lactobacilli, possibly through the activation of the NF-κB signaling pathway [134]. 

Although the data are diverse, they indicate that the microbiota-induced innate and adaptive host immune responses could contribute to HPV persistence and cHSIL development. It is this interplay between HPV-infected host cells, the local immune microenvironment and the vaginal microbiota that could help determine the course of disease.

## 4. Modulation of the Local Microenvironment by Immunotherapy

Non-surgical therapy for cHSIL is warranted especially in the women with a future wish to conceive considering the impact a LLETZ has on subsequent premature birth and subfertility [19,20,21,22]. An alternative therapy modulating the immune microenvironment of cHSIL resulting in immunological eradication of hrHPV infected cells with lesion regression and HPV clearance is an opportunity considering the natural course of a hrHPV infection with a lack of adequate immune responses. Different systemic and local immunotherapies, e.g., topical imiquimod and therapeutic HPV vaccination, are already used in study context and clinical practice. The importance of a preexistent, pro-inflammatory immune infiltrates seems pivotal in the potential of responses upon these immunotherapies, which Abdulrahman et al. have shown to be true for vulvar HSIL [135]. Interestingly, in vulvar HSIL, neither imiquimod nor therapeutic HPV vaccination are able to change the so called ‘cold’ immunosuppressed preexistent microenvironment into a pro-inflammatory, so called ‘hot’, immune inflamed lesion [135,136,137]. 

### 4.1. Imiquimod

Imiquimod is an immune modulating cream which induces a local immune response at the cervix by binding to TLR7 on dendritic cells, monocytes and macrophages. TLR7 activation causes secretion of cytokines, including interferon (IFN)-α and IFN-γ, IL-6, IL-12, and tumor necrosis factor (TNF)-α, resulting in increased antigen presentation by LCs, higher natural killer cell activity and stimulate a type 1 T cell immune response to target HPV infected cells [23,138,139,140,141]. Most importantly, the cytokines stimulate a CD4+ and CD8+ pro-inflammatory T cell immune response by which HPV is cleared [23,138,139,141]. Initially, topical imiquimod was used as a treatment for external anogenital warts, actinic keratoses, and superficial basal cell carcinomas. However, it can be an effective, non-invasive alternative for cervical, vaginal and vulvar HSIL [140]. 

Two randomized controlled trials (RCT) have been conducted to evaluate the effect of imiquimod on cHSIL. Both studies defined regression as LSIL or complete remission of cHSIL after treatment. Grimm et al. performed a double-blind RCT in which a 16 week imiquimod regime was compared to a placebo in 59 women with untreated cHSIL. The regression rate was 73% in the imiquimod group, compared to 39% in the placebo group. Moreover, HPV infection was cleared in 60% of women in the imiquimod group [24]. A more recent RCT, conducted by Fonseca et al., showed a regression rate of 61% after 12 weeks of imiquimod application, compared to 22.5% in the control group [23]. In a recently performed patients’ preference multi-center, non-randomized trial in cHSIL patients in the Netherlands, 61 patients were treated with imiquimod. Imiquimod 5% cream was self-applied by use of a vaginal applicator in a dose of ½ sachet three times a week for in total 16 weeks [27]. Imiquimod was successful in 60% of patients and hrHPV was cleared in 69% of patients; however, 21% of patients discontinued treatment due to side effects [27]. All clinical studies are summarized in Table 3. 

In cHSIL, studies on immune responses in relation to imiquimod efficacy are limited. In vulvar HSIL, studies showed that normalization of immune cell counts in the vulvar microenvironment and a preexisting coordinated local immune response of CD4+ and CD8+ T cells are related to clinical responses to imiquimod treatment [87,135,136,137]. Since TLR7 expression is increased in women who cleared HPV16, imiquimod might contribute to a faster clearance of HPV16 in these women [75,76]. Moreover, if imiquimod is able to overcome the reduced TLR expression in persistent hrHPV infections, it can have a promising role in the treatment of persistent hrHPV infected cHSIL [75]. Therefore, although many patients accomplish a clinical success, imiquimod fails to induce a clinical response in terms of lesion regression in a notable number of patients, which is potentially related to differences in the preexistent immune microenvironment (e.g., ‘hot’ and ‘cold’ pre-treatment immune microenvironment) [23,24,142,143]. 

### 4.2. Therapeutic HPV Vaccination

The purpose of therapeutic vaccination against cervical HSIL and other types of HPV-related malignancies is to augment the T cell response against tumor specific antigens (TSA) created by oncogenic viruses or DNA mutations [147]. E6 and E7 are the primary viral oncoproteins in cHSIL and cervical cancer, thus being excellent TSAs for therapeutic vaccination [148]. Two randomized, placebo-controlled, clinical studies investigating the effects of HPV16 E6/E7 vaccinations in patients with cHSIL reported increased amounts of circulating IFN-γ-producing HPV16 specific T cells; however, no histological changes in cHSIL were observed after vaccination [149,150]. Another study observed a regression rate of 39% after administering a vaccination with HPV L1E7 VLP [151]. Successful treatment of HPV16 and HPV18 induced cHSIL was achieved by Trimble et al., by administering three doses of a therapeutic synthetic DNA vaccine targeting HPV16 and -18 E6 and E7 proteins [152]. Moreover, 40.3% of HPV-16 infections were cleared in the vaccination group, compared to 12.5% in the placebo group. Furthermore, higher numbers of HPV-specific, activated CD8+ T cells were seen in patients with lesion regression. These CD8+ T cells infiltrated both the epithelium and stroma [152]. This is in line with the findings of Maldonado et al., who observed a threefold increased amount of CD8+ T cells in dysplastic epithelial and stromal compartments after therapeutic HPV16 E6/E7 vaccination [153]. As described before, CD8+ infiltration in dysplastic epithelium is associated with regression [85]. The above data suggest that therapeutic vaccinations can boost a HPV-specific pro-inflammatory immune response; however, clinical histologic regression rates need to be improved.

### 4.3. Other Topical Treatments

Cidofovir is an antiviral drug that has been used in both cHSIL as well as in vHSIL with clinical responses in approximately 50% of patients [154,155,156,157,158]. In a RCT for cHSIL, 53 women were randomized for 3 mL 2% cidofovir in Intrasite gel or a placebo 6 weeks before planned conization. Cidofovir induced 60.8% complete responses compared to 20% in the placebo group and viral clearance was increased in the cidofovir group [154]. Correlations of cidofovir and immune responses have not been reported to our knowledge.

Modulation of the VMB from a dysbiotic state to a lactobacillus dominated state could affect the immune microenvironment, thereby aiding spontaneous regression as well as the response to immunotherapy. Existing antimicrobials, probiotics, intravaginally delivered vaginal lactobacilli formulations, novel antimicrobials, biofilm disruptors and vaginal microbiota transplantation (VMT) could be used alone or in combinations to reach homeostasis as reviewed by Laniewski et al., however the effect on the microenvironment and HPV clearance is unknown [129].

A multitude of other topical treatments exist, for example 5-fluorouracil, trans-retinoic acid, interferon, and herbal treatments. These are reviewed in Desravines et al. and Mutombo et al. [159,160].

Immunotherapy in cHSIL may enhance the preexisting immune environment and hereby contribute to lesion regression and therapy effectiveness. The importance of the preexistent lesional immune infiltrates is an important lead to explore effective (immune) biomarkers for patients responsiveness to immunotherapy for cHSIL. Wang et al. recommend immunotherapy for highly immune infiltrated lesions, since so called ‘cold’ immune microenvironments, with low immune infiltrates and low immune-related gene expression have a higher immunologic ignorance, leading to a higher chance of unresponsiveness to immunotherapy, in particular immune checkpoint therapy [56].

## 5. Conclusions

In this review, an overview of the importance of the local immune infiltrates in the microenvironment of cervical neoplasia is given in terms of impact on spontaneous regression and importance of preexistent infiltrates to immunotherapy responses. The local immune infiltrates in spontaneously regressive cLSILs and cHSILs are characterized by both high stromal and epithelial infiltration of CD8+, low infiltration of Tregs and stromal CD138+ B cells, high ratios of epithelial CD4+/CD25+, and high expression of TLR 2, 3, 7–9 [56,83,85]. In the induction of clinical responses to immunotherapy, a preexistent local immune infiltrate seems important. However, overall studies of immune correlates for treatment response are limited and lack an in-depth analysis of distribution, ratio, and function of the innate and adaptive immune cells. Currently, it is not possible to select patients with cHSIL who will regress or progress or will respond to imiquimod cream application or other non-surgical therapies. In order to enable a more personalized management of patients with cHSIL, in-depth studies, including different compartments of the immune microenvironment in combination with clinical parameters, are warranted.

## 6. Author Commentary

Cervical HSIL affects many women worldwide, and next to the concern and anxiety at being confronted with a lesion of malignant potential, cHSIL affects a large group of young women with unfulfilled pregnancy wishes. Although surgery in the form of a LLETZ is effective, this treatment may lead to subsequent subfertility and pregnancy complications. Expectative management for CIN 2 is a save option; however, it requires follow up and prolongs the time of uncertainty where a significant percentage of women will eventually need additional treatment anyhow. Alternative non-surgical treatments for cHSILs are warranted and topical imiquimod cream has proven to be a widely available, effective and save alternative, with clinical success rates of approximately 60%. With high drop-out rates of 20%, due to severe side effects, and being an intensive treatment for 16 weeks, biomarkers identifying patients that will respond to treatment are needed. Biomarkers would be valuable in patient selection, patient counseling, and in preventing unnecessary exposure to side effects in patients with a low chance of response and encouragement to complete treatment for those with a higher chance of responding.

Viral factors and the host’s immune response are key in spontaneous regression since the vast majority of patients is able to clear a cervical HPV infection asymptomatically and is able to mount an HPV-specific immunity. The local immune system in the cervix seems essential in determination of the natural course of the HPV infection, therefore the identification of immune biomarkers for spontaneous regression of CIN 2, or response to immunotherapy in cHSILs are thought to be related to preexistent immune infiltrates in the cHSIL lesion. Current evidence of intralesional immune cells in cHSIL is, however, based on small sample sizes and limited assays, and is generally based on a single stain immunohistochemistry identifying type and number of single immune cell markers. The complex mechanisms and interactions between the various immune cells in the microenvironment response are, therefore, as yet largely unknown. Unraveling of these mechanisms by in-depth comprehensive analysis of both the immune and adaptive immune system and its interactions would aid in elucidating the complex mechanisms involved in the natural history of cervical dysplastic lesions.

This review shows the importance of the host’s intralesional immune response to HPV in the progressive course of cervical dysplasia, in spontaneous regression, as well as in responsiveness to immunotherapies as imiquimod and therapeutic vaccination. Compared to healthy cervix and cLSIL, cHSIL is associated with an increase in iDC and macrophages (especially M2), a decrease in LC, CD4+, and CD8+ T cells and more Tregs and PD-L1+ cells. Wang et al. showed (in limited study numbers) that immune activation and surveillance start at early stages of infection; however, cHSIL seems on the tipping point in the balance of immune activation and suppression [56]. Identifying the key players is of importance and other studies investigating the role of immune cells in persistence and progression of cHSIL lesions, show an association with an anti-inflammatory infiltrate characterized by high numbers of Tregs, M2 macrophages, and cytokine production of IL-10, TGF-β and IDO, which induces T cell anergy [51,52,56,76], while elevated CD8+ T cells and activated CD4+ T cells indicate favorable outcomes.

Prediction of spontaneous regression or patient responses to immunotherapy upon preexistent immune infiltrates, might improve patient selection and upon patient selection, improve effectiveness of the therapy. In several solid tumors, responses to checkpoint inhibitors as anti-PD-L1 are associated with the immune-inflamed phenotypes in terms of immune-inflamed, immune-excluded or immune-deserted [142,143]. A phase II trial (NCT04712851) to evaluate the effect of pembrolizumab, an anti-PD-1 monoclonal antibody, in the treatment of cHSIL is currently recruiting patients [161]. Current evidence clearly hints toward the importance of preexistent intraepithelial infiltrates with T cells, LCs/DCs, macrophages, and low numbers of Tregs, which is a lead for future research. To explore whether imiquimod-induced regression of cHSIL can be predicted by a clinical biomarker, a new study is being initiated (NL79879.100.22), in which an in-depth analysis of the immune microenvironment of cHSIL via multispectral immunofluorescence (mIF) will be performed. mIF is an efficient and reliable technique to monitor the immune microenvironment. In addition to complex cell-to-cell biology, spatial interactions can be studied in great detail [135,162]. Next to immune biomarkers, other possible viral and host factors should be further explored investigating the interaction with the lesion microenvironment and the way the immune system responds to HPV. Host and viral methylation are well studied in various stages of infection forming potential biomarkers for lesion regression or persistence [163,164]. Methylation assay is an interesting method to study different targets concerning cervical carcinogenesis. Increased methylation of these targets can result in gene silencing and thus further progression of cHSIL into cervical cancer [164]. The interaction and correlation with the immune microenvironment however is unknown. A noteworthy technique for monitoring the immune microenvironment are RNA based gene expression patterns, for example by using NanoString technology [165]. Another potential host factor affecting the immune microenvironment is the vaginal microbiome, which shows clear association to HPV infection persistence and can modulate the local immunity [132,166].

Next to identifying patients that will regress spontaneously or respond to immunotherapy, an important clinical research question is related to the cHSIL patients who lack a proper preexistent immune infiltrates, the so called ‘cold’ lesions. Imiquimod and therapeutic vaccination do not seem to be able to turn these ‘cold’ lesions into ‘hot’ lesions. Identifying ways of modulating the intralesional immune infiltrates could result in improved spontaneous regression and responses to imiquimod therapy. Strategies to turn a cold tumor in to a hot tumor are in exploration in malignancies, as recently extensively reviewed by Gerard et al., yet the possibilities in pre-malignancies are unknown [143]. Exploration of additional strategies to enhance the pro-inflammatory microenvironment in cHSIL non-responders are warranted.

In conclusion, future studies should focus on in-depth and broad analyses of the immune microenvironment in relation to the spontaneous regression of cHSIL and in response to immunotherapy in order to identify possible (immune) biomarkers for clinical use. Furthermore, other viral and host processes impacting the immune microenvironment should be explored, in order to find additional therapies able to modulate the immune system in cHSIL patients who do not respond to existing therapies.

## Figures and Tables

**Figure 1 jcm-11-01432-f001:**
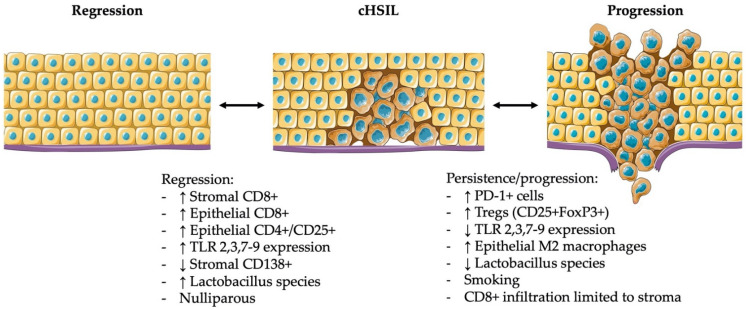
Immunological and clinical factors contributing to the spontaneous regression, persistence, or progression of cHSIL. This figure was created using adapted images of Servier Medical Art, licensed under a Creative Commons Attribution 3.0 Unported License.

**Table 1 jcm-11-01432-t001:** Overview of the treatment guidelines of cLSIL and cHSIL per country/continent.

Country or Continent (Organization) [References]	cLSIL	cHSIL	Current Mode of Treatment
*Australia* *(Cancer Council Australia)* [28]	Observation is preferred, repeat HPV test after 1 year.	CIN 2: - Treatment is recommended. - Women who are concerned about potential effects of treatment on future pregnancy outcomes: observation is acceptable. CIN 3: - Treatment is recommended.	LLETZ, cold knife conization or hysterectomy.
*Europe* *(European Federation of Colposcopy)* [29,30]	Observation is preferred, repeat cytology after 1 year.	Treatment is recommended.	LLETZ, cold knife conization or laser excision.
*Great Britain* *(The British Society for Colposcopy and Cervical Pathology)* [31]	Observation is preferred, repeat cytology after 1 year.	Treatment is recommended.	LLETZ, cold knife conization or laser excision.
*United States of America (American Society for Colposcopy and Cervical Pathology)* [32]	Observation is preferred, repeat cytology after 1 year.	CIN 2: - Women < 25 years: observation is preferred. - Women ≥ 25 years: treatment is recommended. - Women who are concerned about potential effects of treatment on future pregnancy outcomes: observation is acceptable. CIN 3: - Treatment is recommended.	LLETZ, cold knife conization, laser excision, cryotherapy, laser ablation or thermoablation.

cLSIL = cervical low-grade intraepithelial lesion; cHSIL = cervical high-grade intraepithelial lesion; HPV = human papillomavirus; CIN = cervical intraepithelial neoplasia; LLETZ = large loop excision of transformation zone.

**Table 2 jcm-11-01432-t002:** Reported changes in immune microenvironment of cLSIL and cHSIL compared to healthy cervical tissue.

			cLSIL Immune Microenvironment			cHSIL Immune Microenvironment			References
			Overall	Stroma	Epithelium	Overall	Stroma	Epithelium	
Cellular	Pro- inflammatory	iDC	Not reported	Not reported	Not reported	↑	Not reported	Not reported	[45]
		mDC	=	=	=	↓	Not reported	↓	[46,47,48,49,50]
		LC	=	=	=	↓	Not reported	↓	[46,47,48,49,50]
		M1	=	=	=	↑	↑	Not reported	[45,51,52,53]
		CD4+	↓	↓	↓	↓↓	↓↓	↓↓	[37,39,54,55,56]
		CD8+	↓	↓	↓	↓	↓	↓↓	[39,54,55,56]
	Anti- inflammatory	M2	↑	↑	↑	↑↑	↑↑	↑↑	[53]
		Treg	↑	↑	=	↑↑	↑↑	=	[37,45,54,56,57,58]
		PD-1+ T cells	↑	Not reported	Not reported	↑	Not reported	Not reported	[37,45,54,56,57,58]
		PD-L1+ cells	↑	↑	↑	↑↑	Not reported	Not reported	[59,60]
		IDO-1+ cells	=	Not reported	Not reported =	↑	↑	↑	[45,56,61]
Cytokines		TGF-β	↑	Unknown	Unknown	↑	Unknown	Unknown	[55,62,63,64,65]
		IL-10	=	Unknown	Unknown	↑	Unknown	Unknown	[55,62,63,64,65]
		IL-35	Not reported	Unknown	Unknown	↑	Unknown	Unknown	[55,63,64,65]

iDC = immature dendritic cell; mDC = mature dendritic cell; LC = Langerhans cell; M1 = type 1 macrophage; M2 = type 2 macrophage; Treg = regulatory T cell; PD-1 = programmed death-1; PD-L1 = programmed death ligand-1; IDO-1 = immunosuppressive enzyme indoleamine 2,3-dioxygenase-1; TGF-β = transforming growth factor- β; IL = interleukin; ↑ = increase; ↑↑ = strong increase; ↓ = decrease; ↓↓ = strong decrease; = = similar as control.

**Table 3 jcm-11-01432-t003:** Overview of clinical studies evaluating imiquimod for the treatment of cHSIL.

First Author, Year [References]	Treatment	Study Design	Sample Size of Imiquimod Users	Primary Endpoint	Main Clinical Outcome (*n*)
Hendriks, 2022 [27]	Imiquimod vs. LLETZ	PS	61	Clinical regression	Regression: 60% hrHPV clearance: 69%
Fonseca, 2021 [23]	Imiquimod vs. LLETZ	RCT	49	Clinical regression	Regression: 60.5% (23) Persistence: 39.5% (15) Progression: 0% (0)
Cokan, 2021 [26]	Imiquimod vs. LLETZ	RCT	52	Clinical regression	Regression: 62.8% (27)
Chen, 2013 [144]	Imiquimod after surgical treatment	RS	76	HPV clearance	HPV clearance: 76.3% (58)
Grimm, 2012 [24]	Imiquimod vs. placebo	RCT	30	Clinical regression	Regression: 73% (22) Remission: 47% (14)
Lin, 2012 [145]	Imiquimod	PS	1. 26	1. HPV clearance in women with persistent HPV-infection, but normal Pap-smear	1. 69.2% (18)
			2. 72	2. Clinical regression and HPV clearance	2. 51.4% (37)
Pachman, 2011 [146]	Imiquimod vs. excision or ablation	RCT	28	Recurrence within 2 years	Recurrence: 14% (4)

*n* = number of patients; LLETZ = large loop excision of transformation zone; PS = prospective study; HPV = human papillomavirus; RCT = randomized controlled trial; RS = retrospective study.

## Data Availability

Not applicable.

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
