# Peer review of "Importance of the Immune Microenvironment in the Spontaneous Regression of Cervical Squamous Intraepithelial Lesions (cSIL) and Implications for Immunotherapy"

_jcm, 2022, doi:10.3390/jcm11051432_

Round 1

Reviewer 1 Report

Caroline Muntinga et al. submitted a review about the immune microenvironment in the cervical lesions. They well summarized the state of art to underly why the evaluation of immune environment is crucial for prognosis and immunotherapy. It was a pleasure to read and review this manuscript, which was clear and very informative.

However, it needs some minor revisions.

  • In my mind, the title of the review should be modified because the review is about the immune environment of cervical squamous intra-epithelial lesions and not only of “high grade” lesions
  • Table 1 : for European guidelines, the reference date is 2007 but a more recent reference exists (doi: 10.1093/annonc/mdp471)
  • Lines 102-105 : “Since hrHPV infections are primarily localized in the epithelium, the lack of systemic immune responses may be a reflection of a lack of systemic viremia as HPV sheds in the epithelium and exposure of HPV to the immune system is reduced” à this sentence is redundant and should be rephrased
  • Figure 1 : as this figure is really relevant and has a high potential for citations, I suggest a figure edition. This figure deserves to be more eye-catching. The epithelium schemes present a low resolution, they look crooked, and the colors / shapes of the cells are not the same in the 3 schemes.
  • Table 2 :
    • Abbreviations (iDC, mDC, M1, etc.) must be explained
    • The table must include the appropriate references. A last column should be added
    • What the criteria to distinguish “normal” and “further” increases ?
    • What is the difference between “?” and “N/A” ?
  • Line 162-167 : why is PD-L1 mentioned in the text and not reported in Table 2 ?
  • Lines 176-180 : this sentence makes a confusion link between cHSIL and VLP-based vaccine, which is not reported by the cited references. It should be rephrased.
  • Part 5 : I am not familiar with the “expert commentary” in a review. This part does not bring a lot of new data so I don’t get why it is here. Moreover, who is the expert ? The authors ? just one of them ? Another expert ?
  • Lines 481-484 : this sentence suggests that PD-1+ T-cells are increased in cHSIL comparing to cLSIL. The table 2 reported a similar increase. It should be clarified.
  • Among 144 references, 5 are highlighted, and 2/5 are self-citations (ref 17 and 122). Is it fair ?

Reviewer 2 Report

This article must better define the purpose for which it was written. The topic may be of interest but should be less confusing. It is necessary to report what kind of revision it is, the goal and what it adds.
In the natural history of HPV infection I would also add the role of age linked to the immune system. In this regard, I would invite the authors to see the following paper: PMID: 33795132.
What does it mean that you have performed a "systematic search" on pubmed? What guideline did you run for this study? Report in detail following the checklist (PRISM).
The role of the microbiota should also be added.
Reading it, it feels like a narrative review to me.

Reviewer 3 Report

This is a very comprehensive review article and well organized. There is a lot of rich information that has been provided. Here are some suggestions;

  1. Given the density of data provided, it might be useful to add some more sub-headings to the paragraphs to delineate the sections - the authors do provide a good summary para for each current section. There are several facets like immune system categories, types of lesions, and HPV variants. Sub headings will allow the reader to focus a not get lost. For example, in section 1, you can have APC and Macrophages as your sub headers.
  2. You may want to put a bulleted list of key points in each section before going into the supporting information. Rather than a summary para, provide a list up front.
  3. Conclusions should be section 5 - Check numbering
  4. It might be useful to specifically identify the biomarkers of use to monitor the local immune micro-environment given the information you have already synthesized. This will target further investigations and confirmatory studies for researchers. for example - markers listed in table 2 can be used to develop a decision tree in predictions for the use of immunotherapy.
  5. Could you discuss the utility of new tools and techniques that could be used to monitor the immune micro-environment? While that could become an entire section, it merits a paragraph at a minimum.
  6. In the expert commentary section, for the last paragraph that starts on line 516, it might be useful to provide examples of further studies to guide the reader who might be interested in designing experiments.

Reviewer 4 Report

This review is well written and covers important aspects of cHSIL progression and immune mechanisms. The authors describe a hot microenvironment for the progression from low grade lesions to high grade and identify biomarkers like IDO1 and IL10 from the literature survey. These should be definitely validated in a larger patient population.

 Minor comments

Table 1. can you include the current modes of treatment?

 A brief mention of STING pathway in CINs could be added.

The fine line between CIN regression and progression is driven by the viral oncogene expression in the cervical cancer.  However, HPV driven oral/anal cancers are different in that multiple forms of viral episomes exist. Can the authors briefly comment on the differences in tumor microenvironment here? Is there a role for viral tropism here?

Also, the role of DNA damage response leading to CIN progression may be mentioned.

 Please mention Cidofovir, Pembrolizumab in treatment modalities.

There are some unwanted bullet point materials in the references. Remove.
